# Tremor activity inhibited by well-drained conditions above a megathrust

Junichi Nakajima[1] & Akira Hasegawa[2]

Tremor occurs on megathrusts under conditions of near-lithostatic pore-fluid pressures and extremely weakened shear strengths. Although metamorphic reactions in the slab liberate large amounts of fluids, the mechanism for enhancing pore-fluid pressures along the megathrust to near-lithostatic values remains poorly understood. Here we show anti-correlation between low-frequency earthquake (LFE) activity and properties that are markers of the degree of metamorphism above the megathrust, whereby LFEs occur beneath the unmetamorphosed overlying plate but are rare or limited below portions that are metamorphosed. The extent of metamorphism in the overlying plate is likely controlled by along-strike contrasts in permeability. Undrained conditions are required for pore-fluid pressures to be enhanced to near-lithostatic values and for shear strength to reduce sufficiently for LFE generation, whereas well-drained conditions reduce pore-fluid pressures at the megathrust and LFEs no longer occur at the somewhat strengthened megathrust. Our observations suggest that undrained conditions are a key factor for the genesis of LFEs.

[1] Department of Earth and Planetary Sciences, School of Science, Tokyo Institute of Technology, 2-12-1, Ookayama, Meguro-ku, Tokyo 152-8551, Japan. [2] Research Centre for Prediction of Earthquakes and Volcanic Eruptions, Graduate School of Science, Tohoku University, 6-6 Aramaki-Aza-Aoba, Aoba-ku, Sendai 980-8578, Japan. Correspondence and requests for materials should be addressed to J.N. (email: nakajima@geo.titech.ac.jp).

Non-volcanic tremor frequently occurs down-dip of the megathrust seismogenic zone[1,2] in response to small stress fluctuations[3] associated with mechanically weakened shear strengths, which result from extremely high pore-fluid pressures and very low effective normal stresses[4]. Recent laboratory data suggest that a highly permeable serpentinite layer above the megathrust acts to concentrate slab-derived fluids in the region below the mantle-wedge corner, and this process enhances pore-fluid pressures at the megathrust[5]. However, drainage of the megathrust will reduce the amount of fluids there, and it is unclear how the bulk hydrological properties above the megathrust govern pore-fluid pressures at the megathrust. Furthermore, tremor occurs over a wide depth range at the megathrust[6–8], thereby raising the question of whether a particular type of metamorphism at a specific pressure and temperature condition generates tremor activity. Although it is difficult to quantify the permeability contrasts that regulate fluid fluxes at the megathrust, the rate of fluid leakage to the overlying plate may be correlated with the degree of metamorphism above the megathrust. In this case, the amount of fluid channelled along the megathrust should show a negative correlation with the degree of the metamorphism of the overlying plate.

In the Nankai subduction zone, Japan, non-volcanic deep tremor is observed to coincide temporally with short-term slow-slip events (SSEs), and occurs within a limited depth range of 30–35 km over an along-strike length of ∼700 km (refs 1,9), associated with the subduction of the Philippine Sea slab. Although tremor signals are continuous and elusive, isolated pulse-like signals identified as low-frequency earthquakes (LFEs) are often observed within complicated tremor signals[10]. As LFEs coincide spatially with tremor activity[11], here we use the locations of LFEs routinely determined by the Japan Meteorological Agency[12] as a proxy for tremor activity. In Nankai, there are two distinct spatial gaps in LFE and short-term SSE activity; a persistent gap of ∼60 km is observed in the Kii channel (the Kii gap) and a transient gap of ∼30 km occurs in the Ise bay (the Ise gap)[1,9,11]. No, or limited, LFE (that is, tremor) activity occurs beneath Kanto and Kyushu[1,13,14] located at the margin-parallel extension of both sides of the well-defined zone of LFE activity. Therefore, an investigation of the seismic properties of different regions where LFEs are present or absent will yield a better understanding of factors controlling the genesis of LFEs.

In this study, we estimate seismic velocity, attenuation and anisotropy structures in the Nankai subduction zone over an along-strike distance of ∼1,000 km, and find anti-correlation between LFE activity and the degree of metamorphism above the megathrust. We infer that well-drained conditions metamorphose the overlying plate, reduce pore-fluid pressures at the megathrust and consequently inhibit LFE activity. Our observations provide a new perspective toward the understanding of the mechanism for elevating pore-fluid pressures at subduction zones.

## Results

**Seismic velocities and LFE activity.** The observed P-wave (dVp) and S-wave (dVs) velocity perturbations vary significantly along the LFE band (blue outline in Fig. 1a) and show the presence of low-velocity anomalies in the overlying plate beneath Izu–Kanto, the Ise gap, the Kii gap, and Kyushu, which are all regions where LFEs rarely, or never, occur (Fig. 1b,c, Supplementary Fig. 1, and Methods). We calculate dVp, dVs and Vp/Vs above the megathrust along the LFE band and average the values every ∼20 km along strike. This calculation reveals that LFEs do not occur on the megathrust, where dVp and dVs are lower than approximately −4% and Vp/Vs is either lower than ∼1.70 or higher than ∼1.80 (Fig. 2a–c). The average dVp and dVs values are,

respectively, 0.1% ± 3.0% and −0.3% ± 3.4% above areas of LFE activity, and −3.7% ± 1.9% and −5.8% ± 2.8% above areas lacking LFEs (Supplementary Fig. 2). These results suggest a systematic change in seismic velocities in the overlying plate between areas with and without LFE activity.

**Seismic attenuation and anisotropy.** The correlation between P-wave attenuation ($Q_P^{-1}$) and LFE activity is not as clear as that observed for seismic velocities, but LFEs are scarcest, where the megathrust is overlain by highly attenuative areas (that is, with $Q_P^{-1}$ values of approximately >0.006) (Figs 1e and 2d, Methods). Figure 2e shows that delay times between the fast and slow S-wave arrivals derived from shear-wave splitting analysis are large (> ∼0.10 s) for rays propagating in areas lacking LFE activity (that is, Kanto, the Kii gap and Kyushu). The delay times are larger than those observed in the overlying upper crust, which typically has values of 0.035–0.075 s (ref. 15), suggesting the presence of anisotropy in the lower crust or in the mantle wedge above areas without LFE activity. The clear spatial correlation between LFE locations and the seismic velocity, attenuation, and anisotropy anomalies observed in this study demonstrates that the heterogeneous structure of the overlying plate reflects conditions that enhance pore-fluid pressures along the megathrust. Although it is incontrovertible that fluids are released by metamorphic dehydration reactions in the slab[16], the generation of LFEs appears not to be related to the bulk seismic properties in the subducting crust (Supplementary Fig. 3).

## Discussion

Temperature influences seismic properties in the sense that high temperatures lower seismic velocity and enhance seismic attenuation. Therefore, the low-velocity and high-attenuation anomalies observed above the megathrust beneath Kanto cannot be explained by the low-temperature conditions that are inferred from low measured heat fluxes[17] and the deep cutoff depths (25–30 km) of crustal earthquakes[18] (Fig. 2f). Although it is unclear whether temperatures are locally high in the Ise and Kii gaps, the near-constant earthquake cutoff depth of ∼15 km suggests little change in the overlying-plate temperature from Tokai to Shikoku. Compositional variations in the overlying plate may explain the observed variations in seismic properties. However, one of the major tectonic features at Nankai is the intermittent growth of accretionary complexes developed sub-parallel to the arc[19], and this is unlikely to occur solely in discrete geological units in the Ise and Kii gaps, which are oriented transverse to the strike of the accretionary complexes. One hypothesis that might explain the variations in seismic properties along the LFE band is along-strike variations in the degree of prograde metamorphism above the megathrust, which may be proportional to the rate of fluid leakage from the subducting slab to the overlying plate.

The results of thermal-petrologic modelling predict that at Nankai, a large amount of fluid is liberated from the subducting crust at depths of 30–60 km (refs 13,16). Whether or not the slab-derived fluids rise into the overlying plate probably depends on hydrological conditions above the megathrust. If drainage of the megathrust occurs effectively, the overlying plate becomes significantly metamorphosed and seismic velocities decrease regardless of whether the megathrust is contacted with mantle materials or the lower crust[20]. The high-Vp/Vs anomalies (that is, >1.80) observed beneath Kanto and Kyushu can be explained by mantle serpentinization as shown in previous studies[21,22], whereas the low-Vp/Vs values (that is, <1.70) observed in the Kii and Ise gaps may be associated with the existence of fluid-filled pores with an aspect ratio of ∼0.1 (refs 23,24). Increased proportions of silica-rich minerals in the lower crust may also

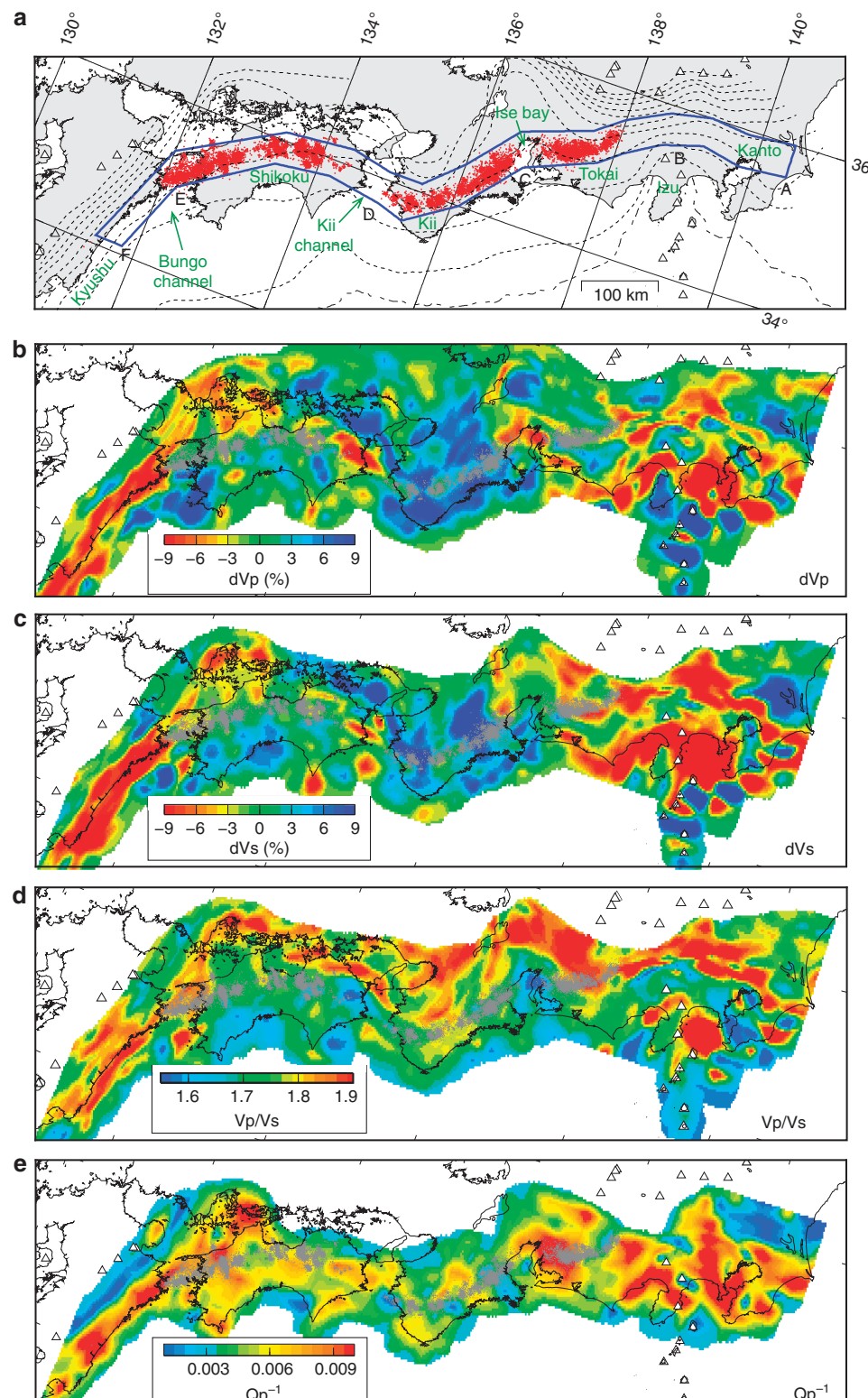

**Figure 1 | Map of low-frequency earthquake activity and seismic properties above the megathrust.** (**a**) Map showing the low-frequency earthquake (LFE) band (outlined in blue). LFEs and active volcanoes are denoted by red dots and white triangles, respectively. Labels A–F mark locations of interest for LFE activity. Location A is used as the origin for the along-strike coordinate for the plots shown in Fig. 2. Names of the regions discussed in the text are also shown. Broken lines denote depth contours of the upper surface of the Philippine Sea slab with an interval of 10 km (refs 43,44). (**b**) Map of P-wave velocity perturbations (dVp), (**c**) S-wave velocity perturbations (dVs), (**d**) Vp/Vs, and (**e**) P-wave attenuation ($Q_p^{-1}$) along a surface 3 km above the Philippine Sea slab. Velocity perturbations are deviations from an average velocity at each depth, and Vp/Vs and $Q_p^{-1}$ are shown in absolute values. LFEs are denoted by grey dots.

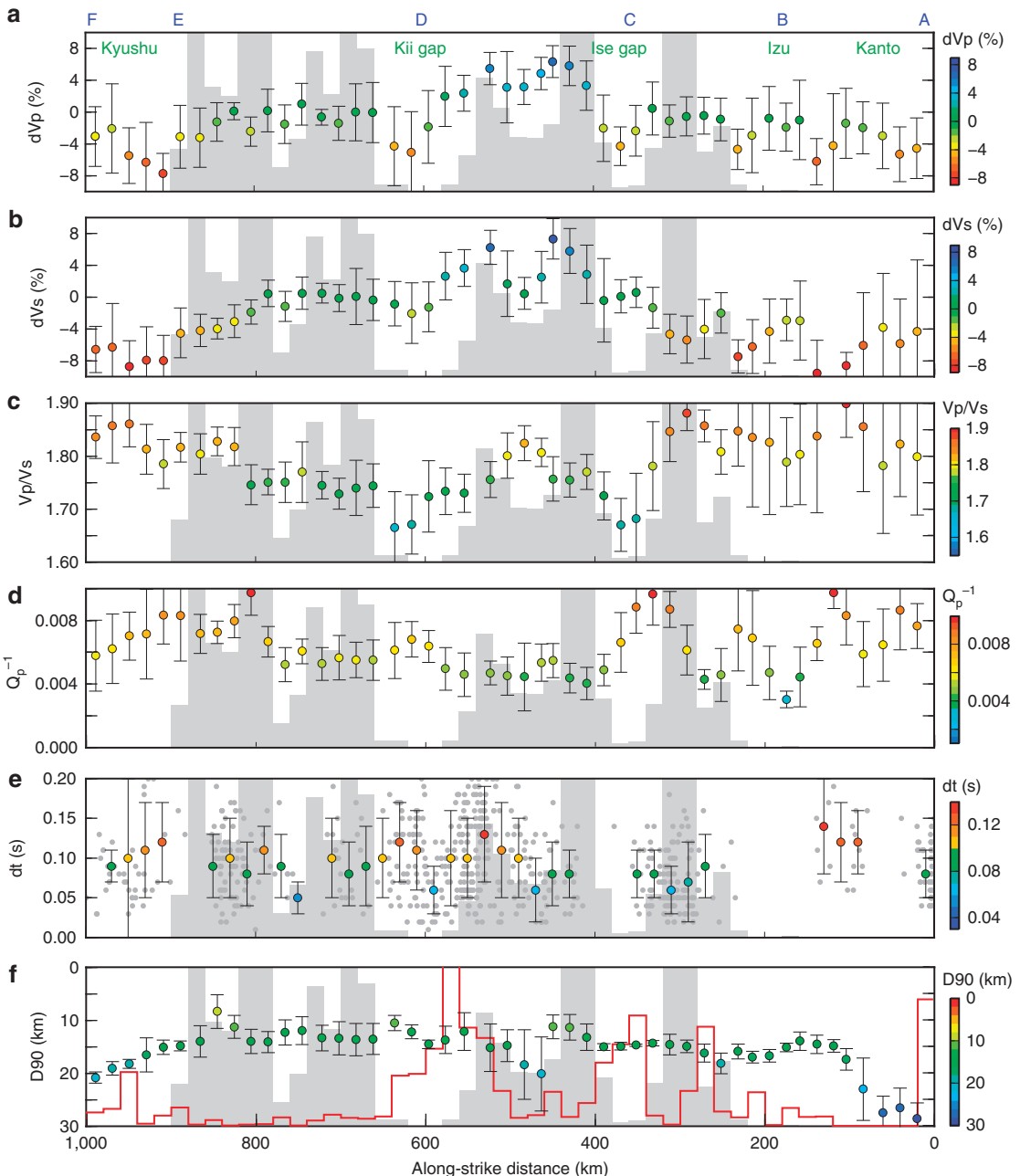

**Figure 2 | Seismic properties above the megathrust compared with low-frequency earthquake activity.** Along-strike variations in: (**a**) P-wave velocity perturbations (dVp); (**b**) S-wave velocity perturbations (dVs); (**c**) Vp/Vs values; and (**d**) P-wave attenuation ($Q_p^{-1}$) averaged every ∼20 km over the layer located 1–4 km above the Philippine Sea slab beneath the low-frequency earthquake (LFE) band shown in Fig. 1a; (**e**) delay times (dt) derived from shear-weave splitting analysis (grey dots denote a delay time measured for each ray path, plotted at the epicentre–station mid-point); and (**f**) D90 (ref. 18), the depth above which 90% of earthquakes in the overlying plate occur. The grey histogram in each panel denotes the number of LFEs (the maximum frequency of 1,000), and the red histogram in **f** shows the number of earthquakes (the maximum frequency of 3,000) that occur in the layer located 10–30 km above the Philippine Sea slab. The vertical bars adjacent to each data point in the panels denote one-sigma uncertainties.

contribute partially to the reduced Vp/Vs values[20]. The high (>1.80) or low (<1.70) Vp/Vs values observed above areas without LFE activity are probably two different manifestations of fluid–rock interactions in the overlying plate. Serpentinization of the mantle or hydration of the lower crust as a result from fluid fluxes from the megathrust would enhance seismic attenuation and anisotropy[25,26], explaining the high attenuation and large anisotropy observed above LFE-devoid areas. Therefore, the along-strike variations in seismic properties revealed in this study suggest that the overlying plate is less metamorphosed above

areas with LFE activity and is significantly metamorphosed beneath Kanto, the Kii and Ise gaps and Kyushu, where no, or limited, tremor (that is, LFE) occurs at depths of 30–35 km (refs 1,14).

We interpret the anti-correlation between LFE activity and the inferred degree of metamorphism above the megathrust as being caused by an along-strike variation in hydrological conditions in the overlying plate. An impermeable overlying plate confines fluids to the megathrust, whereas fluids escape from the megathrust if the overlying plate is permeable.

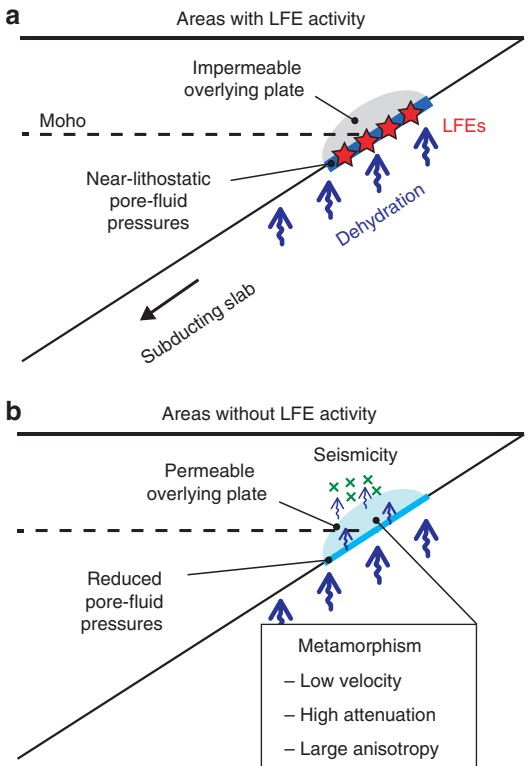

**Figure 3 | Schematic models for the development of pore-fluid pressure along the megathrust.** (**a**) Model of enhanced pore-fluid pressure along the megathrust under undrained conditions. (**b**) Model of reduced pore-fluid pressure along the megathrust and enhanced metamorphism of the overlying plate under well-drained conditions. Low-frequency earthquakes (LFEs) are isolated pulse-like signals with a predominant frequency of ~2 Hz that are observed in continuous tremor signals.

Undrained conditions at the megathrust elevate pore-fluid pressures and result in a low degree of metamorphism in the overlying plate. Pore-fluid pressures that are enhanced to near-lithostatic values under undrained conditions lower the shear strength of the megathrust sufficiently to facilitate repeated short-term SSEs and LFEs[3,4] (Fig. 3a). In contrast, in Kanto, the Ise and Kii gaps, and Kyushu, a portion of the fluids at the megathrust rises into the permeable overlying plate, reducing pore-fluid pressures at the megathrust and metamorphosing the overlying plate. The megathrust becomes somewhat strengthened as a result of the reduced pore-fluid pressures and is no longer weak enough to generate short-term SSEs and LFEs (Fig. 3b). We suggest that the degree of metamorphism above the megathrust is anti-correlated with pore-fluid pressures along the megathrust and can be used as a proxy for the fluid flux from the megathrust, given a spatially constant flux of slab-derived fluids averaged over geological time scales.

The large number of crustal earthquakes in the Kii and Ise gaps (red histogram in Fig. 2f) suggests that LFE activity and seismicity in the overlying plate are anti-correlated. A similar spatial pattern, in which tremor activity is inversely proportional to seismicity, has been reported in Cascadia[27,28], where this anti-correlation is interpreted in terms of a semi-continuous relaxation of strain in the overlying plate produced by frequent episodic tremor and slip[28]. Although strain relaxation by episodic tremor and slip may have some role in Nankai, we consider that the observed spatial anti-correlation of seismicity with LFE activity relates principally to the magnitude of fluid flux from the megathrust. A well-drained megathrust allows over-pressurized

fluids to migrate into the overlying plate, and the occurrence of shallow seismicity is, therefore, facilitated owing to the consequent increase in pore-fluid pressures and decrease in the shear strength of crustal faults[29]. Seismicity in the overlying plate may represent one manifestation of the fluid leakage from the megathrust.

Previous studies suggest that the lithological properties of the overlying plate have a key role in controlling frictional properties along the megathrust. Evidence includes spatial correlations between recurrence intervals of SSEs and geologic terranes[30], between locations of long-term SSEs and LFEs and heterogeneous fluid transport properties in the overlying plate[31], and between segment boundaries of short-term SSEs and areas where seismic properties above the megathrust varies sharply[32]. Our observations add to the growing body of evidence that LFEs occur only under undrained conditions and that a highly metamorphosed, permeable overlying plate inhibits the generation of LFEs. This hypothesis contradicts proposed positive roles of serpentinization in the overlying plate on the enhancement of pore-fluid pressures at the megathrust, in which permeability anisotropy of a serpentinite layer accumulates fluids below the mantle-wedge corner[5] or redundant fluids that are not consumed during serpentinization elevate pore-fluid pressures[33]. We infer that undrained conditions above the megathrust are controlled by the intrinsic lithological structure of the overlying plate, because our observations suggest that the stagnant overlying plate above the locations of LFE is less metamorphosed despite the subduction of the Philippine Sea plate over at least the last 15 Myr (ref. 34). More active mineralogical fluid–rock interactions, such as the influence of silica precipitation on permeability, would elevate pore-fluid pressures locally above the mantle wedge corner[35,36]. However, only this process cannot explain the observations that tremor activity occurs across a wide range of depths at the megathrust, from near the trench[6–8] to down-dip of the seismogenic zone[1,2]. Unless the frictional properties of the megathrust lie in a stable-sliding regime, LFEs could potentially occur in all locations where the megathrust is undrained and weakened.

## Methods

**Travel-time tomography.** Travel-time tomography method[37] was applied to image seismic velocities in the study area, using arrival-time data from 14,458 earthquakes recorded at 778 stations (Supplementary Fig. 4a), with 446,805 P-wave and 392,404 S-wave arrivals. We did not include arrival-time data from LFEs in the analysis. Grid nodes were spaced at intervals of 0.2° (~20 km) horizontally and 5–10 km vertically. A one-dimensional velocity structure derived from the JMA 2001 velocity model[38] was adopted as the initial velocity model, and the continental Conrad and Moho discontinuities were set at constant depths of 16 and 32 km, respectively. The subducting Philippine Sea plate was not prescribed in the model space. The final results were obtained after four iterations of the inversion, which reduced travel-time residuals from 0.26 to 0.17 s for P waves and from 0.33 to 0.23 s for S waves. The velocity perturbations shown in Figs 1b,c and 2a,b are the deviations from the average velocity calculated for each depth.

**Resolution tests for travel-time tomography.** We carried out a checkerboard resolution test (CRT) to ascertain the adequacy of ray coverage for travel-time tomography. In the CRT, positive and negative velocity perturbations of 10% were alternately assigned to grid nodes along both the horizontal and vertical directions, and travel times were calculated to generate synthetic data for this model. Synthetic data were constructed using the same source–receiver geometry as the observations, with random noises corresponding to phase-picking errors (a standard deviation of 0.05 s for P waves and 0.15 s for S waves). We then inverted the synthetic travel-time data using the initial model without any velocity anomalies. We calculated the recovery rate along a curved surface 3 km above the Philippine Sea plate, which is defined as the rate of the inverted (recovered) values with respect to the input (correct) values assigned in the synthetic model. Supplementary Fig. 5a,b shows that the data set can resolve seismic velocity anomalies above the megathrust with a spatial resolution of ~20 km. To test the reliability and robustness of the obtained images, we further conducted three sets of inversions with non-horizontal geometries of the Conrad and Moho discontinuities[39], grid nodes shifted horizontally by a half-grid space, and a

different earthquake data set. Supplementary Fig. 6 shows that a clear correlation between LFE activity and dVp above the megathrust is also obtained by tomographic inversions with the different model descriptions. We also checked the sensitivity of the initial velocity model and grid spacing on obtained tomographic images and confirmed that all the test results indicate the reliability and robustness of velocity images above the megathrust.

**Attenuation tomography.** Seismic attenuation was estimated via a three-step approach[40] that minimizes the potential trade-offs between unknown parameters to be solved, in addition to determining an attenuation term, $t^*$, along a ray path. We first estimated the corner frequencies of earthquakes using the S-coda-spectral-ratio method with a time window of 10 s taken at twice the theoretical S-wave travel time. We then performed a joint inversion to determine $t^*$ and frequency-dependent site-amplification factors for velocity spectra of the direct P waves, with a window length of 2.56 s. An $\omega^2$ source model[41] was assumed throughout, with observed spectra modelled in a frequency range of 1–32 Hz. Finally, we carried out a tomographic inversion to obtain the three-dimensional P-wave attenuation structure in the study area. The number of $t^*$ values used in the inversion was 105,819, measured for 7,383 earthquakes at 625 stations (Supplementary Fig. 4b). Grid-node-spacing in the model space and model descriptions were identical to those used for the travel-time tomography analysis. The three-dimensional P-wave velocity model estimated in the travel-time tomography was used as the velocity model for the attenuation tomography. The final attenuation inversion results were obtained after four iterations; $t^*$ residuals were reduced from 0.019 to 0.015 s.

**Resolution tests for attenuation tomography.** We carried out two sets of CRTs that assign $Q_P^{-1}$ values of 0.007 and 0.001 alternately to one grid node ($\sim 20$ km) (the one-grid model) and two grid nodes ($\sim 40$ km) (the two-grid model) in the horizontal direction. We calculated synthetic $t^*$ values with the same source–receiver geometry as the observations and added normally distributed random noises with a standard deviation of 0.015 s. Then, we inverted the synthetic data using an initial model with a homogeneous $Q_P^{-1}$ of 0.004 and calculated the recovery rate of the CRTs along a curved surface 3 km above the Philippine Sea plate. Supplementary Fig. 5c,d show that the data set can resolve seismic attenuation anomalies above the megathrust with a spatial resolution of 20–40 km.

**Shear-wave splitting analysis.** We applied the cross-correlation method[42] to S-wave waveforms from earthquakes in the subducting Philippine Sea plate beneath the LFE band (the blue outline in Fig. 1a). In the analysis, we measured delay times between fast and slow S waves and fast S-wave polarization directions for seismograms filtered with the pass-band 1–10 Hz. The length of the time window used for the analysis was chosen to be nearly equal to one cycle of the S wave. We restricted our analysis to ray paths with free-surface incident angles of 35° or less, to avoid contamination of particle motions by converted phases. The distributions of 1,047 earthquakes with focal depths of 30–60 km and 105 stations used in this study are shown in Supplementary Fig. 5c. The directions of anisotropy observed at each station are shown in Supplementary Fig. 7.

**Data availability.** The data that support the findings of this study are available from the corresponding author on request.

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

## Acknowledgements

We used arrival-time data of earthquakes reported in the catalogue unified by the Japan Meteorological Agency. Comments from Y. Ito, T. Taira, K. Kuge and S. Kita were very helpful. T. Nishimura and Y. Asano provided us with the locations of short-term SSEs and very LFEs, respectively. Constructive comments by three reviewers improved the manuscript. This work was supported by the Ministry of Education, Culture, Sports, Science and Technology of Japan, under its Observation and Research Program for the Prediction of Earthquakes and Volcanic Eruptions, and by JSPS KAKENHI Grant Numbers JP16H04040 and JP16H06475.

## Author contributions

J.N. performed data processing and tomographic inversions. Both J.N. and A.H. designed this study and contributed to the interpretations and preparation of the manuscript.

## Additional information

**Competing financial interests:** The authors declare no competing financial interests.

**Publisher's note**: 

