## [Peer Review File · Nature Communications]

Reviewer #1 (Remarks to the Author):

Review of "Tremor activity inhibited by metamorphism above a megathrust" by Nakajima and Hasegawa for Nature Communications.

The manuscript describes a seismic investigation of properties in the vicinity of the ETS zone in Japan above the subducting Philippine plate. It is widely recognized that high pore fluid pressures are needed for this mode of slip behavior to take place. Fluid released as a result of dehydration reactions is thought to be an important source enabling these pressures to be reached, and some authors have suggested that metamorphic reactions help to seal fluid escape paths so that high pressures can be maintained. The current paper introduces an interesting twist in which metamorphic activity in discrete along-strike regions above the megathrust in the overlying plate is interpreted as an indication of fluid escape from the shear zone, leading to local strengthening that explains gaps in tremor activity observed to the south beneath Kyushu, in the Kii channel, in Ise Bay and north of Tokai towards Kanto.

The hypothesis put forward here is novel and represents an imaginative advance upon previous efforts that would be of interest to a broad audience. While I applaud the authors for their efforts in constructing their arguments, I am somewhat discouraged that the data they present does not support their case more convincingly. However, I am not a seismologist, and accept that I may be expecting too much from the evidence the authors have so carefully assembled. With that caveat in mind, I offer a few suggestions for the authors to consider as they revise and clarify their presentation.

The title and the concluding paragraph both assert an active role for metamorphism in inhibiting tremor activity. It seems more natural to me to regard the metamorphism the authors identify as a passive indicator of fluid routing. Certainly, seismic interpretations of conditions 3km above the subducting plate are far too distant to play any direct role in any given slip event. Instead, it seems to me that the metamorphism the authors discuss is more naturally viewed as evidence that fluid is escaping the megathrust. Although I concede it is possible that the metamorphism is a sufficiently strong fluid sink to directly influence fluid pressures on the megathrust, this would imply a more restrictive set of physical circumstances that has not been demonstrated and is unnecessary to meet the authors' needs.

I had to reread the opening paragraph several times before I gleaned the authors' intentions. In part this is because of the novel role of metamorphic reactions invoked in the current work. However, I do feel that several small changes could help to clarify matters. In the sentence that continues through line 14, the subordinate clause concerning ETS updip of the seismogenic zone is not relevant to the current work and it is confusing in a couple of regards - I'd suggest jettisoning it both because it reads as though depths of 10-15km are updip of the seismogenic zone, and because none of the current discussion involves this region. On line 17 it is suggested that Nankai ETS is at a constant depth, but the figures clearly demonstrate that there is a range of depths at which ETS is important - better to quote this range. On lines 27 and 28 I was confused by the reference to conventional models, since most conventional models invoke metamorphic dehydration reactions to generate high fluid pressures and only a small subset of recent papers discusses the potential for metamorphic reactions to influence the local permeability. Whereas the current paper is concerned with metamorphic reactions distant from the megathrust itself indicating fluid escape (i.e. up to 3km above), the idea that restrictions to permeability (resulting for example from silica precipitation) might play a role is different in that it requires a more active connection. To this end, readers might be more well conditioned to receive the authors message if they were expecting to learn how 'Tremor activity inhibited by well-drained conditions above a megathrust'.

The grey dots in the lower two panels of Fig. 2 obscure the color contours and make it impossible to see the important patterns of dVp and dVs. The orange outline from panel a might be a better

alternative.

I had difficulty convincing myself of the correlations the authors interpret between tremor activity and seismic anomalies in Fig. 3. For example, dV_p in the Ise gap looks the same as it does to the north, dV_s looks the same in the Kii gap as it does to the south, the V_p/V_s ratios look the same in Kyushu as in the tremor zone immediately north and the same goes for the Izu region and immediately south. Given that all of these data points are shown with just 1-sigma errors, the attenuation data is essentially the same from along-arc distances of 400-800 km, and the delay times are roughly the same along the entire arc.

The final sentence invokes one particular model (ref. 29) to assert a competition between thermal pressurization and dilatant strengthening. I really appreciate that particular reference, but it's only one model amongst many that call upon neither of these two physical processes. More generally, all successful models of slow slip require high pore fluid pressures. I suggest the authors modify the text to emphasize this point, which in any case is much more closely aligned with the rest of the arguments in their manuscript.

There was a typo on line 110. Owing should be owing.

Reviewer #2 (Remarks to the Author):

This paper demonstrates systematic analyses of seismic structures of the Nankai subduction zone, and concludes the occurrence of episodic tremor and slip (ETS) linked to the metamorphism of overlying plate. The data clearly indicate the heterogeneous structure at depth of ~ 30 km along the subducting plate, in which regions of less ETS activity are characterized by relatively low velocity, low V_p/V_s and large seismic anisotropy. The authors interpret these heterogeneous structures caused by the degree of metamorphism in the overlying plate, which controls whether fluids are drained or undrained, and hence pore fluid pressure, at the possible location of ETS. I'm impressed by the detailed structure and the resolution of seismic data, but have a few concerns on the interpretation of data as described below. If the authors can address these points, I really encourage to publication this paper in Nature Communication.

1) Low V_p/V_s regions in the Kii and Ise gap are explained by the precipitation of silica-rich minerals in the overlying plate. However, as discussed by Audet and Burgman (2014), the silica enrichment results in the reduction of permeability, and thereby fluid overpressure beneath the layer of silica deposition, which conflicts to the less activity of ETS in these regions. Since V_p/V_s is also sensitive to the occurrence of aqueous fluids, the low value in the Kii and Ise gap may indicate the drained environments?

2) The degree of metamorphism can be a primarily factor controlling variation of seismic properties in the overlying plate, but what causes such variation are not clear.

Minor

3) V_p/V_s and attenuation are shown as the extended data, but these properties are key for interpretation of heterogeneous structure; the figures are better to include in the original paper.

4) What is the gray bar in Figure 3? Need explanation in the caption.

5) line 109-111: The uppermost oceanic crust shows a markedly high permeability (e.g., Fisher 1998); however, this feature found mainly in the oceanic seafloor might be diminished in the deeply subducted plate due to compaction and/or peeling during subduction.

Reviewer #3 (Remarks to the Author):

The hypothesis here is interesting and well-presented. Many places in the manuscript, though,

hypotheses or premises are presented as though they are definitely true. I provided alternate wordings for some of these cases.

The major claim of the paper is that LFE/tremor activity on the megathrust is related to seismic properties above the megathrust, where little tremor occurs below low-velocity, presumably metamorphosed regions that are presumably associated with a relatively permeable and drained plate interface with lower pore pressure. I am not sure of the degree of novelty of this claim - but it is certainly of interest to others in the field. The claim, although not definitive, is moderately convincing. This is the nature of this topic, in which relationships can be obscured by significant heterogeneity.

Please detail the relationship of this submitted manuscript to the following published work: Kita, S. and M. Matsubara, Seismic attenuation structure associated with episodic tremor and slip zone beneath Shikoku and the Kii peninsula, southwestern Japan, in the Nankai subduction zone, *Journal of Geophysical Research*, v.121 March 2016. It appears to make some claims related to those made here.

An along-strike contrast between where LFE/tremor occurs and where earthquakes occur in the overlying plate was also reported on Vancouver Island (Kao et al, 2009, JGR Figs. 7 and 9). A similar along-dip contrast in southern Cascadia was reported by Boyarko and Brudzinski (2010, JGR, section 3.3 and Fig 1.b).

Line 13-15- Is that really ETS? I am not sure that this shallow tremor should be called ETS. Episodic implies that it recurs repeatedly, and tremor and slip implies that both processes occur - tremor detected seismically and slow slip detected geodetically. Are all three criteria met for any of the referenced observations?

Line 17 'constant depth'? Give the depth range: 28 to 36 km depth range

Line 21 'may be explained if the overlying plate is less permeable'

Line 24 what controls whether the plate is permeable or not?

Line 26 'undrained conditions are a key factor'

Line 28 'a key role' -> 'the key role'

Line 37 There is no general acceptance that the deformation along megathrust is concentrated in a 5 m thick zone. The evidence from drilling into the Tohoku fault plane is at very shallow depth. And some studies speak of damage zones of 10's of meters (e.g., Keren and Kirkpatrick (2016. JGR)

Line 41 'Therefore,' -> 'If so, '

Lines 41-44 are only true if Lines 37-41 are indeed the case.

Line 45 Near the beginning of this paragraph you should mention tremor and its relation to LFEs and state that you will be using LFEs but their spatial distribution is very similar to that of tremor.

Line 46 'occur within an almost constant depth range of 30-35 km'

Line 59 'all regions where LFEs rarely, or never, occur'

Line 69 'most absent' -> 'scarcest'

Line 75 Please insert statement linking anisotropy to possible fluid fluxes.

Line 77-80 The referencing here is a bit confusing. I have the impression that the authors are disagreeing with some premises of Refs. 15 and 16, but more direct statements are needed to clarify.

Line 82 'vice versa' is not the correct term for this.

Line 82-85 Clearer to write: "Therefore, the low velocity and high-attenuation anomalies observed above the megathrust are not explained by the low-temperature conditions beneath Kanto that are inferred from heat-flow data and deep cut-off depths (25-30 km) of crustal earthquakes (Fig. 3f)."

Line 96 I prefer 'a metamorphosed overlying plate would lower seismic velocities.'

Line 105 'indicate' -> 'suggest'

Line 117-120 Something seems to be missing - this explanation is not convincing because it does not explain different frequency characteristics of the slow slip that occurs, that is, since LFEs are thought to be due to small asperities stressed by surrounding slip why is there no LFEs associated with long-term slow slip? This explanation also ignores that long-term SSEs occur somewhat updip in the subduction zone relative to than short-term SSEs.

Line 121 A spatial association between earthquakes and a lack of tremor on the megathrust has been suggested for Cascadia. A contrast between where LFE/tremor occurs and where earthquakes occur in the overlying plate was also reported on Vancouver Island (Kao et al, 2009, JGR Figs. 7 and 9), and in southern Cascadia (Boyarko nad Brudzinski, JGR 2010, Section 3.3 and Fig. 1b).

Line 127 'The hypothesis' -> 'Our hypothesis'

Line 128 I'm not sure 'suppress' is the right word because the absence of high pore pressure doesn't actively suppress the generation of LFEs, it just doesn't promote them.

Line 134 'depends' -> 'may depend'

Line 135 '... dilatancy strengthening. Hence, unless the frictional properties lie in the stable-sliding regime, ETS with various time constants may potentially occur everywhere that the megathrust is weakened and undrained.

Line 141 'was' -> 'were'

Line 341 iterations

Line 361 LEFs -> LFEs

Point-to-point responses to comments by reviewer #1

Some lengthy comments are divided into sub comments and we replied to each sub comment.

We appreciate your fruitful comments on our manuscript. We found the comments are helpful and have revised the manuscript accordingly. Major revisions that we have made are shown in red in the manuscript. We have also corrected grammatical errors and typo and have made minor revisions including changes of expressions and order of sentences and paragraphs.

I had to reread the opening paragraph several times before I gleaned the authors' intentions. In part this is because of the novel role of metamorphic reactions invoked in the current work. However, I do feel that several small changes could help to clarify matters. In the sentence that continues through line 14, the subordinate clause concerning ETS updip of the seismogenic zone is not relevant to the current work and it is confusing in a couple of regards - I'd suggest jettisoning it both because it reads as though depths of 10-15km are updip of the seismogenic zone, and because none of the current discussion involves this region.

We agree with your suggestion and have deleted the sentences concerning ETS up-dip of the seismogenic zone from the abstract.

On line 17 it is suggested that Nankai ETS is at a constant depth, but the figures clearly demonstrate that there is a range of depths at which ETS is important - better to quote this range.

Because of the word limitation (150 words) of the abstract in Nature Communications, we have deleted the detailed description of tremor activity from the abstract. Instead, we have clarified the depth range of tremor in the main text (lines 40–42).

On lines 27 and 28 I was confused by the reference to conventional models, since most conventional models invoke metamorphic dehydration reactions to generate high fluid pressures and only a small subset of recent papers discusses the potential for metamorphic reactions to influence the local permeability. Whereas the current paper is concerned with metamorphic reactions distant from the megathrust itself indicating fluid escape (i.e. up to 3km above), the idea that restrictions to permeability (resulting for example from silica precipitation) might play a role is different in that it requires a more active connection.

We agree with your suggestion. In the abstract (lines 12–14), we have introduced a

conventional idea that a large amount of fluids released by metamorphic reactions in the subducting slab play an important role on enhancing pore-fluid pressures along the megathrust. Furthermore, we have emphasized our hypothesis that the degree of metamorphism in the overlying plate may be anti-correlated with the fluid leakage from the megathrust and that metamorphic reactions distant from the megathrust can be used as an indicator of fluid escaping from the megathrust (lines 105–108, 128–130, and 140–143).

To this end, readers might be more well conditioned to receive the authors message if they were expecting to learn how 'Tremor activity inhibited by well-drained conditions above a megathrust'.

We agree with your opinion and have changed the title of our manuscript to "*Tremor activity inhibited by well-drained conditions above a megathrust*", as you suggested.

The grey dots in the lower two panels of Fig. 2 obscure the color contours and make it impossible to see the important patterns of dVp and dVs. The orange outline from panel a might be a better alternative.

The grey dots may obscure the important patterns of dVp and dVs, but we plotted the dots onto velocity images to show how LFE locations are correlated with heterogeneous structures above the megathrust. In this sense, we have kept showing LFE locations by grey dots, but the size of dots has been reduced so that the dots do not mask the important patterns. If you would ask us to show the heterogeneous structures without grey dots, we could place them in Supplementary Figure.

I had difficulty convincing myself of the correlations the authors interpret between tremor activity and seismic anomalies in Fig. 3. For example, dVp in the Ise gap looks the same as it does to the north, dVs looks the same in the Kii gap as it does to the south, the Vp/Vs ratios look the same in Kyushu as in the tremor zone immediately north and the same goes for the Izu region and immediately south. Given that all of these data points are shown with just 1-sigma errors, the attenuation data is essentially the same from along-arc distances of 400-800 km, and the delay times are roughly the same along the entire arc.

In Figure 3, seismic properties (velocity, attenuation, and anisotropy) as a whole appear to be correlated with LFE activity. However, we find some uncorrelated parts as you pointed out when we look at the figure carefully. This would be a problem that seismologists often face

because of the presence of errors in data sets and the limited resolution of observations. We consider that somewhat smeared correlations between the seismic properties and LFE activity may be due to intrinsic nature of seismological analyses that use noisy data. We believe that the correlation between LFE locations and seismic properties above the megathrust would become clearer when we collect much larger number of ray paths that propagate around LFE locations. Such analyses and other possible subjects that improve spatial resolution of seismological imaging are left open for future studies, which will gain more insights into the controlling factor for the genesis of LFEs.

The final sentence invokes one particular model (ref. 29) to assert a competition between thermal pressurization and dilatant strengthening. I really appreciate that particular reference, but it's only one model amongst many that call upon neither of these two physical processes. More generally, all successful models of slow slip require high pore fluid pressures. I suggest the authors modify the text to emphasize this point, which in any case is much more closely aligned with the rest of the arguments in their manuscript.

In the original manuscript, we referred to a particular model (ref 29: Segall et al., JGR, 115, B12305, doi:10.1029/2010JB007449, 2010) to explain the slip velocity of SSEs and the presence of long-term SSEs in the Kii channel where no tremor occurs. However, since we do not discuss the genesis of long-term SSEs in the revised manuscript, we have deleted the description on the competition between thermal pressurization and dilatant strengthening and modified the text to emphasize the importance of high pore-fluid pressures on the genesis of LFEs.

There was a typo on line 110. Owing should be owing.

Corrected.

Point-to-point responses to comments by reviewer #2

We appreciate your fruitful comments on our manuscript. We found the comments are helpful and have revised the manuscript accordingly. Major revisions that we have made are shown in red in the manuscript. We have also corrected grammatical errors and typo and have made minor revisions including changes of expressions and order of sentences and paragraphs.

1) Low Vp/Vs regions in the Kii and Ise gap are explained by the precipitation of silica-rich minerals in the overlying plate. However, as discussed by Audet and Burgman (2014), the silica enrichment results in the reduction of permeability, and thereby fluid overpressure beneath the layer of silica deposition, which conflicts to the less activity of ETS in these regions. Since Vp/Vs is also sensitive to the occurrence of aqueous fluids, the low value in the Kii and Ise gap may indicate the drained environments?

We agree with your comment that silica enrichment would result in the reduction of permeability and enhancement of pore-fluid pressures along the megathrust, which conflicts to less LFE activity in the Ise bay and Kii channel. According to your suggestion, we have proposed an alternative idea that the presence of aqueous fluids may be a principal cause of the low Vp/Vs values observed above LFE-lacking areas (see lines 116–118). We have also briefly stated that increased proportion of silica-rich minerals may contribute partially to the reduced low Vp/Vs values (lines 118–119).

2) The degree of metamorphism can be a primarily factor controlling variation of seismic properties in the overlying plate, but what causes such variation are not clear.

This is a difficult question to answer. As discussed in the manuscript, the discrete units without tremor activity appear to be located independent of the major geological unit in southwest Japan. A possible explanation is that undrained conditions above the megathrust are controlled by the intrinsic lithological structure of the overlying plate. If the stagnant overlying plate is intrinsically permeable, slab-derived fluids have significantly metamorphosed the overlying plate over a geological time scale. Our observations suggest that the overlying plate is less metamorphosed above the LFE locations despite the subduction of the Philippine Sea over at least the last 15 Myr. Therefore, we have suggested one possibility that an intrinsic impermeable structure of the overlying plate results in along-arc variation in the degree of metamorphism in the overlying plate (lines 165–169).

Minor

3) Vp/Vs and attenuation are shown as the extended data, but these properties are key for interpretation of heterogeneous structure; the figures are better to include in the original paper.

We have moved Vp/Vs and attenuation images from Supplementary Figure to Fig. 2(d) and (e).

4) What is the gray bar in Figure 3? Need explanation in the caption.

The grey bar in each panel in Figure 3 shows the number of LFEs that occurred inside the orange outline in Figure 2a. We have explained the meaning of grey bars in the caption of Figure 3.

5) line 109-111: The uppermost oceanic crust shows a markedly high permeability (e.g., Fisher 1998); however, this feature found mainly in the oceanic seafloor might be diminished in the deeply subducted plate due to compaction and/or peeling during subduction.

We agree with your comment, and have deleted the description of the permeability of the uppermost oceanic crust. Because quantitative comparison of permeability contrasts at the megathrust is beyond the scope of this paper, this revision does not affect the interpretation and conclusions of the manuscript.

Point-to-point responses to comments by reviewer #3

We appreciate your fruitful comments on our manuscript. We found the comments are helpful and have revised the manuscript accordingly. Major revisions that we have made are shown in red in the manuscript. We have also corrected grammatical errors and typo and have made minor revisions including changes of expressions and order of sentences and paragraphs.

Please detail the relationship of this submitted manuscript to the following published work: Kita, S. and M. Matsubara, Seismic attenuation structure associated with episodic tremor and slip zone beneath Shikoku and the Kii peninsula, southwestern Japan, in the Nankai subduction zone, *Journal of Geophysical Research*, v.121 March 2016. It appears to make some claims related to those made here.

Kita and Matsubara (2016) revealed a spatial correlation between areas where seismic attenuation above the megathrust varies abruptly and segment boundaries of short-term SSEs and tremor. However, they do not discuss a correlation of tremor activity with seismic attenuation in the overlying plate, and thus our manuscript adds to new evidence for the role of the overlying plate on the genesis of tremor activity. We have cited Kita and Matsubara (2016) in the first sentence of the last paragraph (lines 156–161), where we introduce three previous studies concluding that the lithological properties of the overlying plate control frictional properties at the megathrust.

An along-strike contrast between where LFE/tremor occurs and where earthquakes occur in the overlying plate was also reported on Vancouver Island (Kao et al, 2009, JGR Figs. 7 and 9). A similar along-dip contrast in southern Cascadia was reported by Boyarko and Brudzinski (2010, JGR, section 3.3 and Fig 1.b).

We consider this suggestion most important. We have referred to the above two papers (Kao et al. 2009, and Boyarko and Brudzinski, 2010) and discussed the anti-correlation between seismicity in the overlying plate and LFE activity. For this discussion, we have added a new paragraph (lines 144–155) and proposed a hypothesis that the well-drained megathrust allows over-pressurized fluids to migrate into the overlying plate and that the occurrence of shallow seismicity is facilitated owing to the consequent decrease in the shear strength of crustal faults.

Line 13-15- Is that really ETS? I am not sure that this shallow tremor should be called ETS.

Episodic implies that it recurs repeatedly, and tremor and slip implies that both processes occur - tremor detected seismically and slow slip detected geodetically. Are all three criteria met for any of the referenced observations?

We agree with you. The description that we regarded shallow tremor as ETS was misleading, and we have deleted the term 'ETS' to represent tremor and slow slip at shallow depths. As we do not discuss slow slip and tremor at shallow depths, this revision does not affect the interpretation and conclusion of the manuscript.

Line 17 'constant depth'? Give the depth range: 28 to 36 km depth range

Because of the word limitation (150 words) of the abstract in Nature Communications, we have deleted the detailed description of tremor activity from the abstract. Instead, we have clarified the depth range of tremor in the main text (lines 40–42).

Line 24 what controls whether the plate is permeable or not?

This is a difficult question to answer. As discussed in the manuscript, the discrete units without tremor activity appear to be located independent of the major geological unit in southwest Japan. A possible explanation is that undrained conditions above the megathrust are controlled by the intrinsic lithological structure of the overlying plate. If the stagnant overlying plate is intrinsically permeable, slab-derived fluids have significantly metamorphosed the overlying plate over a geological time scale. Our observations suggest that the overlying plate is less metamorphosed above the LFE locations despite the subduction of the Philippine Sea over at least the last 15 Myr. Therefore, we have suggested one possibility that an intrinsic impermeable structure of the overlying plate may result in along-arc variation in the degree of metamorphism in the overlying plate (lines 165–169).

Line 26 'undrained conditions are a key factor'

Line 28 'a key role' -> 'the key role'

This sentence has been deleted during the revision.

Line 37 There is no general acceptance that the deformation along megathrust is concentrated in a 5 m thick zone. The evidence from drilling into the Tohoku fault plane is at very shallow depth. And some studies speak of damage zones of 10's of meters (e.g., Keren and Kirkpatrick (2016. JGR)

Thank you for your comment. As the thickness of the deformation zone along the megathrust is not essential in this manuscript, we have deleted the description on the thickness of the formation zone from the revised manuscript.

Line 41 'Therefore,' -> 'If so, '

Lines 41-44 are only true if Lines 37-41 are indeed the case.

This sentence has been deleted during the revision.

Line 45 Near the beginning of this paragraph you should mention tremor and its relation to LFEs and state that you will be using LFEs but their spatial distribution is very similar to that of tremor.

This comment is very important, and we have introduced the relation between tremor and LFEs in terms of waveforms and locations. The sentences we have added are as follows (lines 42–46). *"Although tremor signals are continuous and elusive, isolated pulse-like signals identified as low-frequency earthquakes (LFEs) are often observed within complicated tremor signals¹⁰. As LFEs coincide spatially with tremor activity¹¹, here we use the locations of LFEs routinely determined by the Japan Meteorological Agency¹² as a proxy for tremor activity."*

Line 46 'occur within a almost constant depth range of 30-35 km'

This statement has been deleted during the revision.

Line 59 'all regions where LFEs rarely, or never, occur'

Line 69 'most absent' -> 'scarcest'

Corrected.

Line 75 Please insert statement linking anisotropy to possible fluid fluxes.

We have stated the linking of anisotropy and fluid fluxes from the slab in lines 121–124.

Line 77-80 The referencing here is a bit confusing. I have the impression that the authors are disagreeing with some premises of Refs. 15 and 16, but more direct statements are needed to clarify.

We have revised the context and corrected the citation of references.

Line 82 'vice versa' is not the correct term for this.

We have deleted the term 'vice versa'.

Line 82-85 Clearer to write: "Therefore, the low velocity and high-attenuation anomalies observed above the megathrust are not explained by the low-temperature conditions beneath Kanto that are inferred from heat-flow data and deep cut-off depths (25-30 km) of crustal earthquakes (Fig. 3f)."

Corrected (lines 95–98).

Line 96 I prefer 'a metamorphosed overlying plate would lower seismic velocities.'

This sentence was modified during the revision.

Line 105 'indicate' -> 'suggest'

Corrected. (line 125)

Line 117-120 Something seems to be missing - this explanation is not convincing because it does not explain different frequency characteristics of the slow slip that occurs, that is, since LFEs are thought to be due to small asperities stressed by surrounding slip why is there no LFEs associated with long-term slow slip? This explanation also ignores that long-term SSEs occur somewhat updip in the subduction zone relative to than short-term SSEs.

We agree with your comment. In the original manuscript, we referred to a particular model (ref 29: Segall et al., JGR, 115, B12305, doi:10.1029/2010JB007449, 2010) to explain the slip velocity of SSEs and the presence of long-term SSEs in the Kii channel where no tremor occurs. Since we do not discuss the genesis of long-term SSEs in the revised manuscript, we have deleted the description on the competition between thermal pressurization and dilatant strengthening and revised the texts to emphasize the important role of high pore-fluid pressures on the generation of LFEs. This revision follows the comment by reviewer #1.

Line 121 A spatial association between earthquakes and a lack of tremor on the megathrust has been suggested for Cascadia. A contrast between where LFE/tremor occurs and where earthquakes occur in the overlying plate was also reported on Vancouver Island (Kao et al, 2009, JGR Figs. 7 and 9), and in southern Cascadia (Boyarko and Brudzinski, JGR 2010, Section 3.3 and Fig. 1b).

We consider that this comment is very important. It is really worthwhile describing that we observe an anti-correlation between LFE activity and seismicity in the overlying plate in both Cascadia and SW Japan. We have discussed this issue in the second paragraph from the last (see our reply to your 2nd comment for details)

Line 127 'The hypothesis' -> 'Our hypothesis'

This sentence was modified during the revision.

Line 128 I'm not sure 'suppress' is the right word because the absence of high pore pressure doesn't actively suppress the generation of LFEs, it just doesn't promote them.

Line 134 'depends' -> 'may depend'

These sentences were changed or deleted during the revision.

Line 135 '... dilatancy strengthening. Hence, unless the frictional properties lie in the stable-sliding regime, ETS with various time constants may potentially occur everywhere that the megathrust is weakened and undrained.

Corrected.

Line 141 'was' -> 'were'

Line 341 iterations

Line 361 LEFs -> LFEs

Corrected.

Reviewer #1 (Remarks to the Author):

The authors have made substantial revisions that address the points raised in my earlier review.

There remain a few places where statements might be misinterpreted, but these are relatively minor objections. For example, on line 88, the statement that slab heterogeneity is unimportant is meant to refer to the kind and scale of heterogeneity that is observable with seismic techniques. Some models of ETS rely upon heterogeneity in frictional properties to produce tremor from rate-weakening asperities in a rate-strengthening background.

The manuscript represents a substantial and novel contribution that will be of interest to the broader community and I recommend it be published more-or-less as is.

Reviewer #2 (Remarks to the Author):

I have no more objection.

Reviewer #3 (Remarks to the Author):

Overall, I think the manuscript is almost ready for publication. I listed a few suggestions and remaining questions below.

In general, there are still some spots where inferences are presented as statements of fact. Perhaps the NCOMMS Editor can work with the authors to clarify this.

Line 16 – “properties that are markers of the degree of metamorphism...”

Line 18 – “The extent of metamorphism in the overlying plate is likely controlled by along-strike contrasts in permeability.”

Line 21 – “sufficiently” or “enough”

Line 23 - “undrained conditions are a key factor”

Line 112 – “is” -> “becomes”

Lines 114-121 – so you expect tremor/LFEs only where V_p/V_s is intermediate in value?

Line 120 – “are probably two different manifestations”

Line 128 – “the inferred degree”

Line 146 – whereby -> in which

Line 162 – “and that a highly-metamorphosed, permeable overlying plate”

Line 163-165 – What is the mechanism suggested in the studies cited here (Refs 5,21,33)? Also, it seems that there are several geometries in which permeability could exert a relevant control - there is the impermeability of the overlying crust in bulk, but there is also the possibility of a thin layer of highly impermeable material within or just above the megathrust. Similarly, it is actually spatial variations of permeability that allow pore pressure to build up. A uniformly highly permeable upper plate might not facilitate an increase in pore-pressure as suggested in Lines 153-154.

Line 175-177 – None of the results in this manuscript necessarily pertain to strike-skip environments. Therefore, I suggest that you omit the last sentence.

Point-to-point responses to comments by reviewer #1

The authors have made substantial revisions that address the points raised in my earlier review. There remain a few places where statements might be misinterpreted, but these are relatively minor objections. For example, on line 88, the statement that slab heterogeneity is unimportant is meant to refer to the kind and scale of heterogeneity that is observable with seismic techniques. Some models of ETS rely upon heterogeneity in frictional properties to produce tremor from rate-weakening asperities in a rate-strengthening background.

We agree with your comment, and the description in line 88 was misleading. As the results stated in the line were derived from travel-time tomography, we have clarified that *'the generation of LFEs appears not to be related to the bulk seismic properties in the subducting crust'* (lines 110–111)

The manuscript represents a substantial and novel contribution that will be of interest to the broader community and I recommend it be published more-or-less as is.

We appreciate your fruitful comments on our manuscript during review process.

Point-to-point responses to comments by reviewer #3

Overall, I think the manuscript is almost ready for publication. I listed a few suggestions and remaining questions below. In general, there are still some spots where inferences are presented as statements of fact. Perhaps the NCOMMS Editor can work with the authors to clarify this.

We appreciate your fruitful comments on our manuscript during review process. We have considered your comments listed below and prepared the final manuscript

Line 16 properties that are markers of the degree of metamorphism

Line 18 The extent of metamorphism in the overlying plate is likely controlled by along-strike contrasts in permeability.

Line 21 sufficiently or enough

Line 23 undrained conditions are a key factor

Line 112 is -> becomes

Revised

Lines 114-121 so you expect tremor/LFEs only where V_p/V_s is intermediate in value?

Yes, we expect that tremor/LFEs occur where V_p/V_s above the megathrust has intermediate values (1.7–1.8).

Line 120 are probably two different manifestations

Line 128 the inferred degree

Line 146 whereby -> in which

Line 162 and that a highly-metamorphosed, permeable overlying plate

Revised

Line 163-165 What is the mechanism suggested in the studies cited here (Refs 5,21,33)?

Also, it seems that there are several geometries in which permeability could exert a relevant control - there is the impermeability of the overlying crust in bulk, but there is also the possibility of a thin layer of highly impermeable material within or just above the megathrust. Similarly, it is actually spatial variations of permeability that allow pore pressure to build up. A uniformly highly permeable upper plate might not facilitate an increase in pore-pressure as suggested in Lines 153-154.

This comment is important, and the description suggested above was not accurate. The references cited here claim that a serpentinite layer plays a role on the enhancement of pore-fluid pressures on the megathrust, but the mechanisms considered in the papers are different each other. Thus, we have revised the sentence to explain proposed models more concretely, as follows. *'This hypothesis contradicts proposed positive roles of serpentinitization in the overlying plate on the enhancement of pore-fluid pressures along the megathrust, in which permeability anisotropy of a serpentinite layer accumulates fluids at the mantle-wedge corner⁵ or redundant fluids that are not consumed during serpentinitization elevate pore-fluid pressures³³.'*

Line 175-177 None of the results in this manuscript necessarily pertain to strike-skip environments. Therefore, I suggest that you omit the last sentence.

Deleted as suggested.